# Controlling protein activity by dynamic recruitment on a supramolecular polymer platform

Sjors P.W. Wijnands [1], Wouter Engelen [1,2], René P.M. Lafleur [1], E.W. Meijer [1] & Maarten Merkx [1,2]

Nature uses dynamic molecular platforms for the recruitment of weakly associating proteins into higher-order assemblies to achieve spatiotemporal control of signal transduction. Nanostructures that emulate this dynamic behavior require features such as plasticity, specificity and reversibility. Here we introduce a synthetic protein recruitment platform that combines the dynamics of supramolecular polymers with the programmability offered by DNA-mediated protein recruitment. Assembly of benzene-1,3,5-tricarboxamide (BTA) derivatives functionalized with a 10-nucleotide receptor strand into µm-long supramolecular BTA polymers is remarkably robust, even with high contents of DNA-functionalized BTA monomers and associated proteins. Specific recruitment of DNA-conjugated proteins on the supramolecular polymer results in a 1000-fold increase in protein complex formation, while at the same time enabling their rapid exchange along the BTA polymer. Our results establish supramolecular BTA polymers as a generic protein recruitment platform and demonstrate how assembly of protein complexes along the supramolecular polymer allows efficient and dynamic control of protein activity.

[1] Institute for Complex Molecular Systems, Eindhoven University of Technology, P.O. Box 513, Eindhoven 5600 MB, The Netherlands. [2] Laboratory of Chemical Biology, Department of Biomedical Engineering, Eindhoven University of Technology, P.O. Box 513, Eindhoven 5600 MB, The Netherlands. Sjors P. W. Wijnands and Wouter Engelen contributed equally to this work. Correspondence and requests for materials should be addressed to E.W.M. (email: e.w.meijer@tue.nl) or to M.M. (email: m.merkx@tue.nl)

Adaptive nanostructures based on supramolecular self-assembly are crucial constituents of life. Natural supramolecular protein polymers such as actin filaments and microtubules provide structural integrity and mobility to the cell, whereas cellular membranes allow for compartmentalization of cellular processes[1]. Moreover, signaling pathways that govern processes such as cytoskeletal remodeling, proliferation, gene expression, and metabolic regulation all involve the transient formation of higher-order protein complexes. These signal transduction cascades use specific scaffold proteins, membrane surfaces, or micro-phase-separated states as dynamic recruitment platforms to engage weakly associating proteins into higher-order assemblies[2–4]. By modulating the effective local concentration of proteins rather than their absolute concentration, supramolecular protein recruitment provides an efficient mechanism to gain spatiotemporal control over protein activity in the highly crowded environment of the cytoplasm[5,6]. Moreover, the intrinsic plasticity of supramolecular organization allows reversible remodeling and adaptation to the requirements of a specific signaling pathway[7–9].

The unique properties of natural supramolecular protein assemblies has inspired scientists to create (semi-)synthetic systems that capture characteristic features of their natural counterparts, including (supramolecular) polymers, vesicles, and DNA-origami structures[10–13]. For example, synthetic vesicles have been used to encapsulate or reversibly recruit proteins and their substrates to increase their local concentrations, providing control over reaction specificity and kinetics[11]. One-dimensional templates consisting of covalent copolymers of styrene sulfonate and methyl methacrylate with poly(ethylene glycol) side chains have been used as heparin mimics by decorating them with basic fibroblast growth factors, hereby significantly increasing the stability of the growth factor while applying therapeutically relevant forms of stress[14]. More recently, Stupp and coworkers reported the development of sulfated glycopeptide nanostructures as highly active, adaptive scaffolds for the binding of various growth factor proteins, efficiently promoting bone regeneration in the spine[15]. Another noticeable example is the work of Brunsveld and coworkers, who explored the use of supramolecular polymers consisting of amphiphilic discotics to control the assembly of fluorescent proteins, using Förster resonance energy transfer (FRET) to study the dynamic exchange of protein functionalized monomers between different polymer chains[16,17]. Synthetic platforms based on DNA origami enable excellent control over the spatial orientation and stoichiometry of the recruited proteins, allowing the assembly of protein cascades with increased pathway specificity and turnover rates[13,18–20]. While the attachment of protein-DNA conjugates on DNA-origami scaffolds is in principle reversible through DNA strand exchange, the origami scaffold itself is highly static, hampering the dynamic rearrangement of recruited proteins to, e.g., accommodate mismatches in receptor distances and assembly defects[21,22]. Systems that combine the programmability of DNA hybridization with the self-assembly properties of synthetic polymers have received increased attention, but primarily with the aim of developing materials with tunable properties[13,23,24]. Thus far it has remained challenging to develop synthetic protein recruitment systems that combine the reversibility and orthogonality offered by DNA-based nanotechnology with the dynamics found in natural systems[25–27].

Here we introduce a protein recruitment platform that combines the dynamics of supramolecular polymers with the programmability and orthogonality of DNA hybridization. Our platform is based on the well-characterized supramolecular polymerization of water-soluble benzene-1,3,5-tricarboxamide derivatives (BTA)[28]. Using a combination of threefold hydrogen bonding and hydrophobic interactions, these BTA monomers form micrometer-long one-dimensional polymers in water that allow the incorporation and homogenous exchange of various (functional) BTA monomers[29–31]. Supramolecular assembly allows straightforward tuning of the density of functional BTA monomers within these BTA polymers and previous work has also shown that the distribution of functional monomers can be externally controlled, e.g., via interaction with multivalent templates[32]. To allow specific recruitment of multiple proteins on the BTA polymer scaffold, we synthesized a BTA monomer functionalized with a 10-nucleotide single-strand DNA handle (BTA-DNA), which serves as a receptor in the supramolecular polymer. By employing the programmable and reversible nature of oligonucleotide hybridization, protein-DNA conjugates can be recruited on the BTA polymer with high specificity and temporal control. Our results establish supramolecular BTA polymers as a remarkably robust platform for DNA-mediated recruitment of proteins and demonstrate how the assembly of protein complexes along the supramolecular polymer allows efficient and dynamic control of protein activity.

## Results

**DNA-directed recruitment of proteins on BTA polymers**. Figure 1 shows the principle of DNA-directed protein recruitment on BTA polymers. BTA monomers were functionalized with a 10-nucleotide handle strand (BTA-DNA) by conjugation of an alkyne-functionalized oligonucleotide (5′-GTAACGACTC$_{alkyne}$-3′) to a mono-azide BTA via a copper catalyzed cycloaddition reaction (Fig. 1b). To study specific recruitment of proteins to the BTA polymers, we used the enzyme TEM1-β-lactamase and its inhibitor protein (BLIP), a well-characterized enzyme-inhibitor pair whose affinity is readily tunable by the introduction of various point mutations[33]. β-lactamase and BLIP were both conjugated to a unique 21-nucleotide DNA handle using thiol-maleimide chemistry[34]. To recruit the proteins on the supramolecular BTA polymer, two recruiter strands were designed to be complementary to both the sequence of the DNA handle on BTA-DNA and to those on the enzyme ($R_E$) or the inhibitor ($R_I$). Hybridization of the protein- and BTA handle strands to the recruiter strand should result in the selective recruitment of the respective protein on the supramolecular polymer via formation of a nicked dsDNA duplex. This system thus allows reversible control over supramolecular protein assembly at three different levels: (1) the ratio of BTA-DNA and inert BTA (BTA-3OH), which controls the density of BTA-DNA within the BTA polymer, (2) the concentration of recruiter strands, and (3) the concentration of the DNA-protein conjugates.

**Structural characterization using cryo-TEM and STORM**. The functionalization of BTA monomers with oligonucleotides and proteins significantly changes their physiochemical properties, which may affect their supramolecular assembly and exchange behavior. Cryogenic transmission electron microscopy (cryo-TEM) was therefore used to study the effects of DNA-conjugation and subsequent protein recruitment on the structural properties of the BTA polymers. Figure 2a shows that even in the presence of 25% BTA-DNA, micrometer-long fibrous structures were formed that closely resemble the fibers formed by 100% BTA-3OH[28]. Moreover, similar fibers were also observed upon addition of the two recruiter strands and DNA-functionalized β-lactamase and BLIP (Fig. 2b, c). These results confirm that the directional assembly of BTAs is maintained upon incorporation of BTA-DNA monomers, both in the absence and presence of recruited proteins. When only the recruiter strands were added to the BTA polymers, also tubular structures were observed (Supplementary Fig. 3). It is unclear why these structures were formed

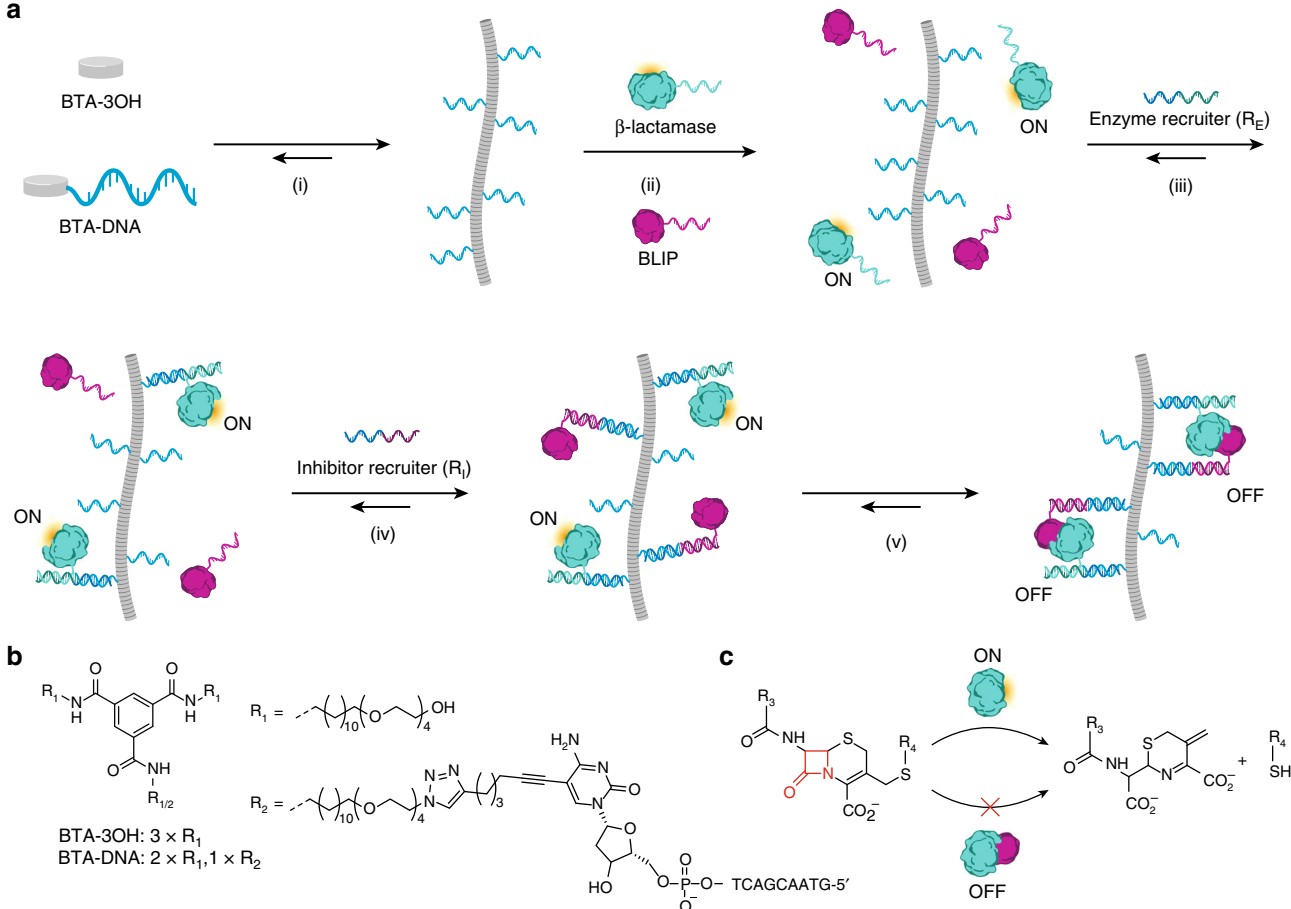

**Fig. 1** Recruitment of β-lactamase and BLIP on a supramolecular BTA polymer via DNA hybridization. **a** Schematic representation of the selective recruitment of proteins to BTA polymers. (i) BTA monomers functionalized with a 10-nucleotide handle strand (BTA-DNA) are co-assembled in a desired ratio with inert BTA monomers (BTA-3OH) to yield supramolecular polymers decorated with DNA receptors. (ii) To allow sequence specific oligonucleotide directed recruitment of the enzyme (β-lactamase) and its inhibitor protein (BLIP), each protein is functionalized with a unique 21-nucleotide handle strand. (iii) Addition of a specific recruiter oligonucleotide that is complementary to the handle strand on the BTA polymer and the enzyme (enzyme recruiter ($R_E$)), results in selective recruitment of the enzyme to the supramolecular polymer. (iv) Addition of the inhibitor recruiter strand ($R_I$) results in the recruitment of the inhibitor to the supramolecular platform. (v) The dynamic nature of the platform allows the rearrangement of the proteins along the polymer to allow enzyme-inhibitor complex formation, resulting in decreased enzyme activity. **b** The chemical structures of BTA-3OH and BTA-DNA. **c** The hydrolysis of a β-lactam-containing substrate which is catalyzed by β-lactamase

in addition to the one-dimensional fibers, but we have observed similar structures when studying charged BTAs.

While cryo-TEM clearly shows that supramolecular polymers are formed in the presence of both the recruiter strands and proteins, these experiments do not provide direct evidence for successful recruitment of the proteins on the supramolecular scaffold. Stochastic optical reconstruction microscopy (STORM) has proven to be a powerful technique to study both the structure and dynamics of a variety of supramolecular polymers, including BTAs[29,35,36]. Here, STORM was used to study the recruitment of the proteins to the BTA polymers. Cy3-labeled BTA monomers (BTA-Cy3) were incorporated in the DNA-decorated supramolecular polymers to visualize the BTA backbone, while β-lactamase and BLIP were labeled with the dyes Atto488 and Cy5 respectively. The labeled proteins and respective recruiter strands were mixed with the BTA polymers and annealed on glass slides by physical adsorption. Multi-color STORM was performed by consecutive excitation of the three dyes and analysis in separate channels. In agreement with the cryo-TEM data, micrometer-long fibrous structures were observed in all three channels (Fig. 2d). These results confirm the successful co-assembly of BTA-DNA and BTA-3OH and subsequent DNA-

hybridization-driven recruitment of both proteins to the polymers, demonstrating the ability of the supramolecular BTA polymers to assemble large, water-soluble structures while conserving their fibrous morphology.

**BTA-templated enzyme-inhibitor complex formation.** Having established the successful recruitment of both the enzyme and the inhibitor protein on the BTA polymers, we next explored whether the recruitment on the supramolecular platform promotes their interaction. To this end, the E104D mutant of β-lactamase was used, which binds to wild-type BLIP with an inhibition constant ($K_i$) of 1.5 μM[37]. This affinity was chosen such that no enzyme-inhibitor complexes are formed when both proteins are present in solution at low nanomolar concentrations. The extend of BTA-templated complex formation, and thus enzyme inhibition, was quantified by monitoring the hydrolysis of a β-lactam-containing fluorescent substrate (CCF2-FA, Fig. 1c). First, the enzymatic activity of 1 nM β-lactamase recruited on the BTA platform (25% BTA-DNA and 20 nM $R_E$) in the absence of inhibitor protein was determined (Fig. 3a). Subsequently, substantial inhibition of enzymatic activity was observed upon addition of 10 nM BLIP

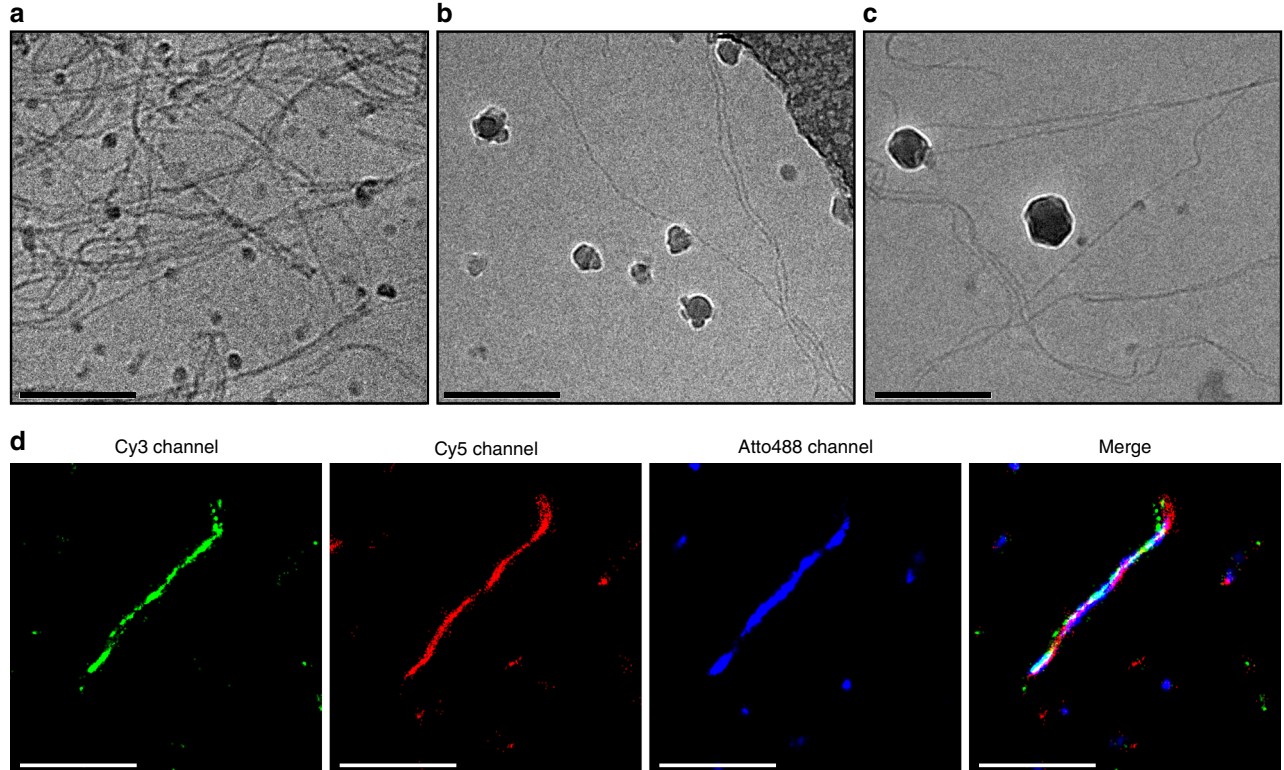

**Fig. 2** Structural characterization of DNA-decorated BTA polymers and recruitment of proteins. Cryo-TEM images of DNA-decorated BTA polymers in PBS (50 mM sodium phosphate, 500 mM NaCl, pH 7.0) with **a** 150 μM BTA-3OH and 50 μM BTA-DNA (25% BTA-DNA), **b** 195 μM BTA-3OH and 5 μM BTA-DNA (2.5% BTA-DNA), 2 μM $R_E$ and $R_I$, and **c** 195 μM BTA-3OH and 5 μM BTA-DNA (2.5% BTA-DNA), 2 μM $R_E$ and $R_I$, 10 nM β-lactamase and 100 nM BLIP. The scale bars represent 200 nm. The dark spherical objects present in the images are crystalline ice particles. **d** Stochastic optical reconstruction microscopy imaging of BTA polymers in TE/Mg$^{2+}$ (10 mM Tris-HCl, 1 mM EDTA, 12.5 mM MgCl$_2$, pH 8.0) with 1.4 μM BTA-3OH, 0.5 μM BTA-DNA and 0.1 μM BTA-Cy3 (70% BTA-3OH, 25% BTA-DNA and 5% BTA-Cy3), 100 nM $R_E$ and $R_I$, 100 nM Atto488-labeled β-lactamase and 7.5 nM Cy5-labeled BLIP. The individual dyes were imaged sequentially and reconstructed in separate channels. The scale bars represent 2 μm

and 20 nM of recruiter strand $R_I$, consistent with efficient formation of the enzyme-inhibitor complex as a result of the recruitment of both proteins on the supramolecular polymer. Importantly, no inhibition of enzyme activity was observed when either one or both of the recruiter strands, or the BTA polymers were omitted (Fig. 3b). Additionally, no inhibition of enzyme activity was observed when only the 10-nucleotide handle strand was added, either in absence or presence of BTA polymers consisting of 100% BTA-3OH (Supplementary Fig. 4). These results proof that the formation of the enzyme-inhibitor complex critically depends on the presence of both the BTA polymers containing BTA-DNA as well as the specific recruitment of each of the two proteins on the supramolecular platform.

To quantify the increase in effective concentration resulting from recruitment of the proteins on the BTA polymer, the enzyme activity was evaluated as a function of inhibitor concentration. To this end, 1 nM β-lactamase was recruited on the BTA polymer together with increasing concentrations of BLIP (0.5–100 nM). Figure 2c shows that upon increasing inhibitor concentrations, the enzyme activity rapidly decreases reaching a maximal enzyme inhibition of ~75% at saturating inhibitor concentrations. Fitting the enzyme activity as function of inhibitor concentration (Eq. 1 in the Methods section) yielded an apparent inhibition constant ($K_{i,app}$) of 2.3 nM, which is 3 orders of magnitude lower than the $K_i$ observed for the enzyme-inhibitor pair in solution. The residual enzyme activity observed at saturating BLIP concentrations is due to the fact that under these experimental conditions hybridization between the enzyme

recruiter strand and BTA-DNA is not complete, leaving 25% of the enzyme free in solution (Supplementary Note 1).

To assess whether the increase in apparent affinity can be understood by a higher local concentration of the proteins upon recruitment on the supramolecular polymer, we calculated the effective concentration of the inhibitor protein using a model that describes the BTA polymer as a cylinder with a radius of 15 nm (Supplementary Note 2 and Supplementary Fig. 5). Assuming that the proteins can reside anywhere within the volume of this cylinder, the model predicts an effective inhibitor concentration of 26 μM, when using a bulk concentration of 10 nM BLIP and 2 μM BTA containing 25% BTA-DNA. This 2600-fold increase in effective concentration corresponds to the increase of the apparent affinity observed in Fig. 3c. The model predicts that the effective inhibitor concentration is determined by the ratio of the amount of inhibitor protein and the total amount of BTA monomers. To test this prediction, BTA polymers were assembled by mixing a constant BTA-DNA concentration of 0.5 μM with BTA-3OH concentrations ranging between 0 to 199.5 μM, yielding BTA polymers containing 0.25 to 100% BTA-DNA. Figure 3d shows that the enzyme activity is clearly dependent on the amount of BTA-3OH present in the polymers. While inhibition remains constant between 25 and 1% BTA-DNA, enzymatic activity increases when the amount of BTA-DNA decreases below 0.75%. At this percentage the model shows that the inhibitor protein is diluted to an effective concentration of 0.8 μM, a concentration that is below the inhibition constant of the enzyme-inhibitor pair (Supplementary Note 2 and

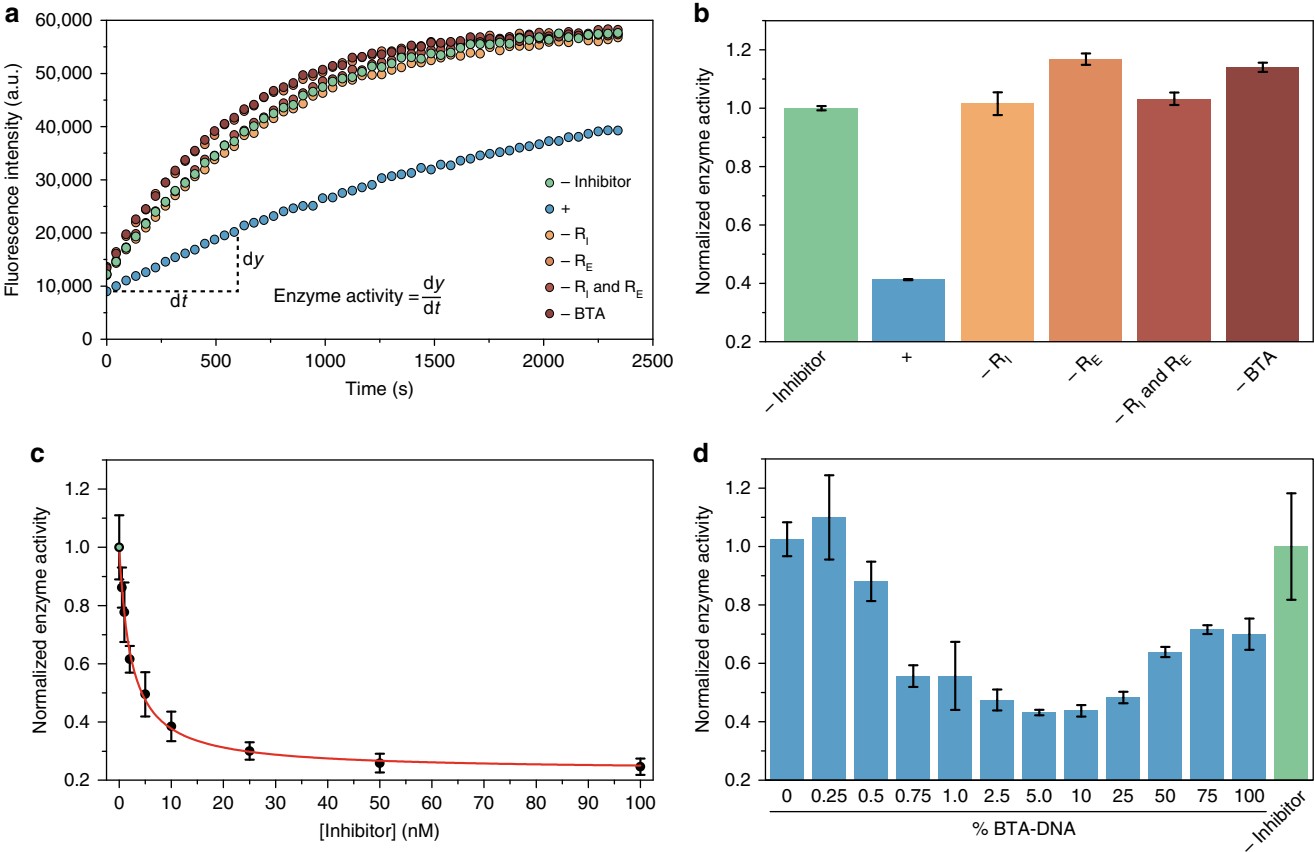

**Fig. 3** Characterization of BTA-templated enzyme-inhibitor complex formation. **a**, **b** Substrate conversion as a function of time by the recruited enzyme in the absence of BLIP (-inhibitor), the fully assembled system (+) and controls lacking one or both of the recruiter strands (-$R_I$, -$R_E$ and -$R_I$ and -$R_E$, respectively) or the BTA polymers (-BTA). Turnover of a fluorescent substrate (CCF2-FA, 2 μM) was monitored by measuring the fluorescence intensity at 447 nm. Experiments were performed using 25% BTA-DNA (0.5 μM BTA-DNA, 1.5 μM BTA-3OH), 20 nM $R_E$ and $R_I$, 1 nM β-lactamase and 10 nM BLIP. Enzyme activities **b** were obtained from the kinetic traces shown in **a** by deriving the slope of the initial increase in fluorescence. Enzyme activities were normalized to a control omitting the inhibitor protein. Error bars represent SEM of duplicate measurements. **c** Normalized enzyme activity as a function of inhibitor concentration (black dots). The enzymatic activities were fitted to Eq. 1, which was derived using the Michaelis–Menten model for competitive inhibition, yielding an apparent inhibition constant ($K_{i,app}$) of 2.3 ± 0.2 nM (red line). Experiments were performed using 25% BTA-DNA (0.5 μM BTA-DNA, 1.5 μM BTA-3OH), 20 nM $R_E$ and 200 nM $R_I$, 1 nM β-lactamase, and 0–100 nM BLIP. Error bars represent s.d. calculated from triplicate measurements. **d** Normalized enzyme activities as a function of BTA-DNA receptor density. Polymers with receptor densities between 100% and 0.25% BTA-DNA were obtained by assembling a fixed concentration of BTA-DNA (0.5 μM) with varying concentrations of BTA-3OH (0–199.5 μM). Experiments were performed using 20 nM $R_E$ and $R_I$, 1 nM β-lactamase and 10 nM BLIP. Error bars represent s.d. calculated from triplicate measurements

Supplementary Table 2). Overall, the effective concentrations calculated using the model correspond very well with the observed amount of enzyme inhibition (Supplementary Fig. 6). Remarkably, the enzyme activity also increases at BTA-DNA densities above 25%. This effect is not due to destabilization of the BTA fibers at high densities of BTA-DNA, as total internal reflection fluorescence microscopy (TIRF) showed the presence of similar fibrous structures over a wide range of BTA-DNA densities, even at 100% BTA-DNA (Supplementary Fig. 7). We therefore attribute the inefficient formation of enzyme-inhibitor complexes to steric hindrance caused by the high density of DNA handles on the BTA polymers above 25% BTA-DNA.

**Dynamics of protein complex assembly.** The results presented above show that the increase in apparent affinity can be quantitatively understood by the higher effective concentration of the enzyme and inhibitor proteins when recruited on the BTA polymer. While this provides a satisfying thermodynamic explanation, the model assumes that a protein can freely diffuse within the cylindrical confinement of the BTA fiber. With the experimental conditions used in Fig. 3a, b, one can calculate that the

average distance between two proteins would be ~85 nm when the recruited proteins distribute homogenously along the BTA polymers (Supplementary Note 3). Since this distance is too large to allow efficient complex formation, it is clear that the dynamic character of the system should allow redistribution of the recruited proteins along the BTA polymer. To gain more insight into the molecular origin of this redistribution, we studied the kinetics of complex formation in three different experiments. In the first experiment, we started with enzyme-inhibitor complexes assembled on BTA polymers containing 5% BTA-DNA and subsequently added an excess of BTA polymers containing 100% BTA-3OH, decreasing the overall BTA-DNA content to 0.25%. According to the results shown in Fig. 3d, this dilution of inhibitor protein over a larger volume of BTA polymers should result in a disruption of the enzyme-inhibitor complex and an increase in enzymatic activity. This experiment thus probes the redistribution of BTA-DNA between BTA polymers and should inform about the kinetics of BTA monomer exchange. Figure 4a shows the measured enzyme activity as a function of time after addition of the inert BTA polymers, revealing a gradual increase in activity over the course of 5 h. The kinetics observed here

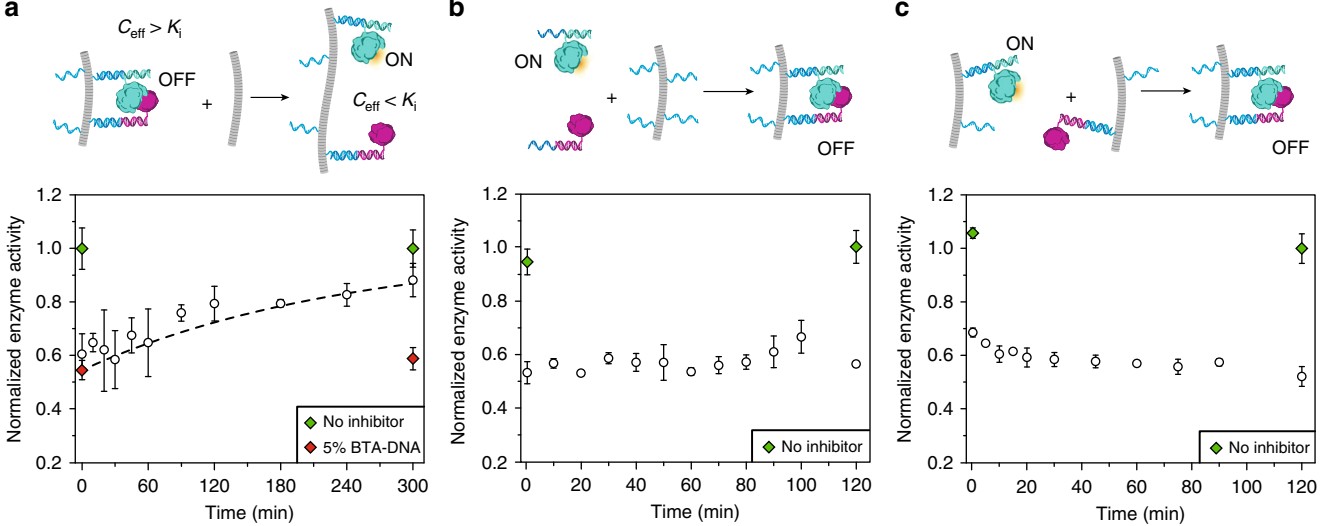

**Fig. 4** Characterization of the dynamics of enzyme-inhibitor recruitment on the supramolecular polymer. **a** Enzyme activity as a function of time after the addition of pre-assembled polymers containing 100% BTA-3OH (9.5 μL, 1 mM) to polymers composed of 5% BTA-DNA (0.5 μM BTA-DNA, 9.5 μM BTA-3OH), 20 nM $R_E$ and $R_I$, 1 nM β-lactamase and 10 nM BLIP (total volume = 40.5 μL), yielding a final BTA-DNA fraction of 0.25% (0.5 μM BTA-DNA, 199.5 μM BTA-3OH, final total volume = 50 μL). The dashed line represents the fitting of a single exponential with fixed values at $t = 0$ of the activity in the 5% BTA-DNA reference sample and at $t = \infty$ of the activity in the reference sample omitting BLIP yielding $t_{1/2}$ = ~3 h. The green diamonds represent the activity of a sample without inhibitor. The red diamonds represent the activity of the system containing 5% BTA-DNA, which was not diluted by addition of 100% BTA-3OH. **b** Kinetics of enzyme-inhibitor complex formation upon recruitment to the DNA-decorated polymers. Enzyme activity was measured at different time-point after adding the BTA polymers to a mixture of β-lactamase, BLIP $R_E$ and $R_I$. Experiments were performed using 25% BTA-DNA (0.5 μM BTA-DNA, 1.5 μM BTA-3OH), 20 nM $R_E$ and $R_I$, 1 nM β-lactamase and 10 nM BLIP. **c** Enzyme activity as a function of time after mixing pre-assembled polymers containing either enzyme and $R_E$ or inhibitor and $R_I$. Experiments were performed using final concentrations of 25% BTA-DNA (0.5 μM BTA-DNA, 1.5 μM BTA-3OH), 20 nM $R_E$ and $R_I$, 1 nM β-lactamase and 10 nM BLIP. Enzyme activities were normalized to a control omitting the inhibitor protein (green diamonds). Error bars represent the s.d. calculated from triplicate measurements

closely resemble those previously determined for the exchange of (functional) BTAs, suggesting that DNA-functionalization does not significantly affect BTA exchange dynamics[30–32].

In the second kinetic experiment, we monitored the rate of enzyme-inhibitor complex formation upon recruitment of the proteins on the supramolecular polymers. In this experiment enzyme activity was monitored over time after adding pre-assembled DNA-decorated BTA polymers to a mixture of β-lactamase, BLIP, $R_E$, and $R_I$. Figure 4b shows that protein recruitment and subsequent formation of the enzyme-inhibitor complex is remarkably fast, reaching equilibrium within minutes. As the observed kinetics are faster than the time scale of BTA monomer exchange, we conclude that this rapid protein complex formation is not controlled by the exchange of BTA-DNA monomers. Instead, we hypothesize that the efficient reorganization of proteins along the BTA polymer is enhanced by rapid association and dissociation of the recruiter strands with/from the BTA-DNA handle strands. A dissociation rate of 0.2 s⁻¹ has been reported for DNA duplexes consisting of 10 base pairs, which is thus consistent with such a mechanism[38].

To probe the kinetics of protein exchange between BTA fibers, a third experiment was done where we prepared separate BTA polymers that were functionalized with either the enzyme or the inhibitor protein (Fig. 4c). Rapid equilibration of proteins was observed between the BTA polymers as the decrease in enzyme activity was complete within 5 min after mixing of the two BTA assemblies. Again, the fast enzyme-inhibitor complex formation cannot be explained by exchange of BTA monomers between fibers, but is likely mediated by rapid, reversible hybridization of the recruiter strands to the handle strands on the supramolecular polymers. Additionally, in this case we can also not exclude the possibility of multivalent interactions between multiple enzymes on one fiber with multiple inhibitors on another fiber.

**Reversible protein recruitment via DNA exchange reactions.** Toehold-mediated strand displacement plays a central role in dynamic DNA nanosystems such as DNA-based actuators and molecular computers[39–41]. Since protein assembly on the BTA scaffold is based on DNA hybridization, we explored whether toehold-mediated strand displacement could be used to reversibly recruit proteins on the BTA polymers. To this end, the enzyme recruiter strand was redesigned to include a 10-nucleotide over-hang ($R_{ET}$). Addition of $R_{ET}$ to BTA polymers pre-incubated with $R_I$, BLIP, and β-lactamase induced the expected decrease in enzyme activity by promoting the formation of an enzyme-inhibitor complex. Next, a twofold excess of a displacer strand (D) was added, which was designed to be fully complementary to $R_{ET}$. Addition of the displacer strand resulted in the complete restoration of enzyme activity, consistent with dissociation of β-lactamase from the BTA polymer (Fig. 5). Figure 5b shows that sequential addition of increasing concentrations of $R_{ET}$ and D resulted in cycling between high and low enzyme activity, proving that the reversible recruitment of β-lactamase can be performed multiple times. The attenuation of reversibility observed after several cycles is likely caused by the high concentrations of recruiter and displacer-strands that may result in alternative association or folding of the toehold on $R_{ET}$, making it unavailable for strand displacement[42].

## Discussion

In this work, orthogonal and reversible recruitment of proteins on a synthetic supramolecular polymer was demonstrated. Our finding that supramolecular interactions between the relatively small BTAs are sufficient to coordinate the assembly of protein complexes via DNA hybridization is quite remarkable. The robust nature of this supramolecular platform is reflected by the observation that the characteristic μm-sized supramolecular polymer

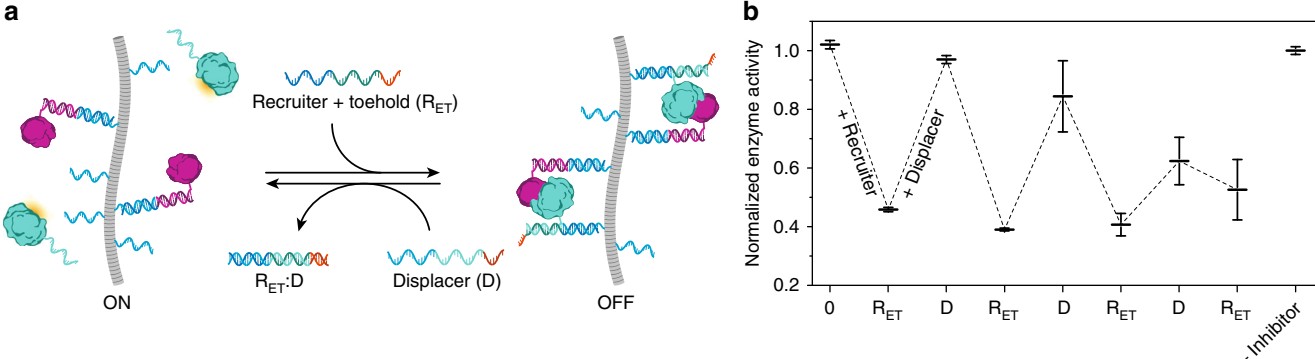

**Fig. 5** Reversible control of enzyme recruitment via DNA strand exchange. **a** The enzyme is recruited to polymers pre-assembled with inhibitor protein by the addition of recruiter strand $R_{ET}$. A 10-nucleotide overhang (toehold) appended to the recruiter strand ($R_{ET}$) allows its displacement by a fully complementary displacer strand (D) via toehold-mediated strand displacement. **b** Normalized enzyme activity upon the sequential addition of recruiter and displacer oligonucleotides. Each iteration, a twofold molar excess of either recruiter or displacer strand is added with respect to the previous stage. Enzyme activities were normalized to a control without inhibitor protein. Experiments were performed using 25% BTA-DNA (0.5 µM BTA-DNA, 1.5 µM BTA-3OH), 20 nM $R_I$, 1 nM β-lactamase and 10 nM BLIP, 0 – 340 nM $R_E$ and 0-168 nM D. Error bars represent s.d. of triplicate measurements

chains are maintained even with extremely high contents of DNA-functionalized BTAs. DNA-mediated assembly of proteins on the BTA scaffold resulted in a large increase in local protein concentration, effectively enhancing complex formation between the reporter enzyme β-lactamase and its inhibitor protein BLIP by 3 orders of magnitude. BTA-templated protein complex formation was also found to be surprisingly fast, reaching equilibrium within minutes. Since exchange of BTA monomers between polymer chains was found to require multiple hours, protein complex formation on the BTA scaffold probably results from rapid association and dissociation of DNA duplexes along the polymer, possibly accelerated by the high local density of DNA handles. The 10-nucleotide DNA-receptor strand used in this study thus provides sufficient thermodynamic driving force to ensure efficient recruitment of low nanomolar concentrations of proteins on the BTA polymer, while at the same time allowing rapid exchange of DNA-conjugated proteins along the BTA scaffold. A more thorough understanding of the importance of supramolecular polymer dynamics and the kinetics of DNA hybridization may be obtained by tuning the length of the DNA receptor strand, and/or by changing the exchange kinetics of the BTA monomer.

While we restricted ourselves in this study to the co-assembly of two proteins to allow straightforward read-out of protein activity, the number of different proteins that can be recruited is limited only by the physical size of the BTA polymer chain, allowing the assembly and characterization of much more complex signaling cascades. A unique feature of the BTA-DNA platform is the dynamic nature of the complexes, enabling the efficient formation of transient complexes without prior knowledge over the exact structure and stoichiometry of such complex signal transduction cascades. Moreover, the supramolecular nature of the BTA platform allows one to efficiently tune the density of DNA handles on the BTA polymer, which in turn allows precise control of the effective local concentration and optimization of the system for different protein complexes. The introduction of DNA-mediated protein recruitment should also allow the integration of supramolecular protein assembly with various forms of dynamic DNA nanotechnology such as DNA-based molecular circuits that allow logic operations, amplification, thresholding, and feedback control. In addition to these fundamental applications, dynamic recruitment of proteins on BTA polymers could also be used for the development of responsive biomaterials and dynamic multivalent ligands, employing molecular recognition based on the presence of multiple target receptors[43]. Finally, we believe that BTA-DNA scaffolds may be applied to control intracellular signaling pathways, as efficient RNA transfection with BTA polymers has already been demonstrated in mammalian cell lines[44]. Once inside the cell, their integration with intracellular pathways could involve the interaction of proteins, aptamers, and small molecules pre-assembled on the BTA-DNA scaffold with messenger RNA, regulatory RNAs, and proteins endogenous to the cell.

## Methods

**General.** Unless stated otherwise, reagents and solvents were purchased from commercial sources and used as received. The alkyne-functionalized oligonucleotide (5′-GTAACGACTC$_{alkyne}$-3′) was obtained from Base Click GmbH. Other oligonucleotides were obtained by high pressure liquid chromatography (HPLC)-purified from Integrated DNA Technologies (oligonucleotide sequences can be found in Supplementary Table 1). Oligonucleotides were dissolved in TE buffer (10 mM Tris-HCl, 1 mM EDTA, pH 8.0) to an approximate concentration of 100 µM as suggested by the manufacturer. Subsequently the concentration was determined by ultraviolet (UV)-visible (NanoDrop 1000, Thermo Scientific), using the extinction coefficients at 260 nm calculated with the DNA UV spectrum predictor tool (Integrated DNA Technologies[45]). Finally, the oligonucleotide stock solutions were diluted in TE buffer to a final concentration of 50 µM and stored at −30 °C.

**Synthesis of BTA monomers.** The chemical structures of the BTA monomers can be found in Supplementary Fig. 1. The synthesis of BTA-3OH has been reported by Leenders and coworkers[28]. The synthesis of BTA-$N_3$ and BTA-Cy3 has been reported by Albertazzi et al.[29]. BTA-(3′-CTCAGCAATG-5′) (BTA-DNA) was synthesized by charging a two-neck flask (10 mL) with a CuSO$_4$ (0.006 mmol, 0.9 mg) solution in H$_2$O (0.2 mL), followed by BimPy$_2$ ligand (0.005 mmol, 1.7 mg) dissolved in DMF (0.5 mL)[29]. To the resultant green mixture, a solution of sodium ascorbate (0.05 mmol, 10 mg) in H$_2$O (0.3 mL) was added, followed by a spoon tip of solid sodium ascorbate, turning the color of the solution yellow. BTA-$N_3$ (0.04 mmol, 50.5 mg) was dissolved in DMF (2 mL) and mixed with a solution of 5′-GTAACGACTC-alkyne-3′ (0.025 mmol, 83 mg) in water (2 mL) producing a fine dispersion. This dispersion was carefully added to the yellow reaction mixture resulting in a total reaction volume of 6 mL after rinsing with additional H$_2$O (0.5 mL) and DMF (0.5 mL). The reaction mixture was stirred at room temperature, and over time, gradually became clear. After 6.5 h, EDTA solution in water was added (1.8 mL, 0.04 mmol) resulting in a color change from bright yellow to deep yellow. Subsequently, BTA-DNA was purified from remaining BTA-$N_3$ by dialysis (molecular weight cut-off (MWCO) = 1 kD, Spectra/Por 7, Spectrumlabs) against water, after which the material was dried by lyophilization, yielding BTA-DNA as a fluffy white material. BTA-DNA was analyzed using mass spectrometry by flow injection analysis on a LCQ Fleet (Thermo Finnigan) ion-trap mass spectrometer in negative mode (Supplementary Fig. 2). BTA-DNA was dissolved at 10 µM in 1:1 isopropanol/water + 1% triethylamine (pH 10) of which 5 µL was directly injected. The mass-to-charge spectrum was deconvoluted using MagTran software. Starting material BTA-$N_3$: calculated MW = 1313.81 g mol$^{-1}$. BTA-DNA: calculated MW = 4429.99 g mol$^{-1}$, observed deconvoluted MW = 4430.5 g mol$^{-1}$. Complex of BTA-$N_3$ and BTA-DNA: calculated MW = 5743.80 g mol$^{-1}$, observed deconvoluted MW = 5742.1 g mol$^{-1}$.

**Synthesis and labeling of β-lactamase-DNA and BLIP-DNA.** The mutagenesis and expression of β-Lactamase and BLIP and conjugation to oligonucleotides have been reported by Janssen et al.[34] For the labeling of the β-lactamase- and BLIP-DNA conjugates for STORM imaging, the proteins were buffer exchanged to PBS (100 mM sodium phosphate, 150 mM NaCl, pH 7.2) using Zeba spin desalting columns (7k MWCO, 0.5 mL, Thermo Fisher). The resulting protein concentrations were determined by measuring the absorbance at 260 nm (NanoDrop 1000, Thermo Scientific) and using the extinction coefficient of the handle oligonucleotides conjugated to the proteins (β-lactamase = $20.39 \times 10^4$ $M^{-1}$ $cm^{-1}$, BLIP = $22.54 \times 10^4$ $M^{-1}$ $cm^{-1}$). To the solutions of β-lactamase and BLIP, 20 molar equivalents of respectively Atto488-NHS (Atto-TEC) and Cy5-NHS (Lumiprobe) from stock solutions in DMSO were added (final DMSO content = 2%) and the reactions were shaken at 850 r.p.m. for 1 h at room temperature. Finally, the labeled proteins were purified from unreacted fluorophores by gel filtration using Zeba spin desalting columns. The average labeling efficiency of the proteins was determined by measuring the absorbance at 260 nm for the proteins and at 500 and 646 nm for Atto488 and Cy5 respectively, given extinction coefficients of the protein-DNA conjugates and the fluorophores (β-lactamase = $20.39 \times 10^4$ $M^{-1}$ $cm^{-1}$, BLIP = $22.54 \times 10^4$ $M^{-1}$ $cm^{-1}$, Atto488 = $9 \times 10^4$ $M^{-1}$ $cm^{-1}$, Cy5 = $25 \times 10^4$ $M^{-1}$ $cm^{-1}$). This yielded average labeling efficiencies of 4.2 labels per β-lactamase and 1.02 label per BLIP.

**Assembly of BTA polymers.** From a stock solution of BTA-3OH in methanol, an appropriate volume was added into a glass vial and dried under vacuum. An appropriate amount of MilliQ water and stock solution of BTA-DNA (500 µM) in MilliQ water was added after which the mixture was stirred at 80 °C for 15 min. The mixtures were then vortexed for 15 s and left to equilibrate at room temperature overnight. For the incorporation of BTA-Cy3 into the polymers, appropriate volumes of BTA-3OH and BTA-Cy3 from stock solutions in methanol were mixed and dried after which the protocol described above was continued.

**Cryogenic transmission electron microscopy.** Samples were vitrified using a computer controlled vitrification robot (FEI Vitrobot™ Mark III, FEI Company). Quantifoil grids (R 2/2, Quantifoil Micro Tools GmbH) were surface plasma treated with a Cressington 208 carbon coater. Vitrified films were transferred into the vacuum of a Tecnai Sphera microscope with a Gatan 626 cryoholder. The microscope is equipped with a $LaB_6$ filament that was operated at 200 kV, and a bottom mounted $1024 \times 1024$ Gatan charged-coupled device (CCD) camera. Co-assemblies of BTA-3OH and BTA-DNA were prepared as stated above at a total BTA concentration of 1 mM with 25% BTA-DNA. After overnight equilibration, these were diluted to 200 µM in PBS for cryo-TEM analysis. The effect of the addition of the recruiter strands and subsequently the recruitment of the proteins on the morphology of the BTA polymers was studied by assembling polymers containing 2.5% BTA-DNA. After overnight equilibration these were diluted to 200 µM in PBS with 2 µM $R_E$ and $R_I$ and additionally with 100 nM BLIP and 10 nM β-lactamase for the latter. Vitrified films containing the BTA polymers were prepared in the Vitrobot that was operated at 22 °C, and at a relative humidity of 100%. In the preparation chamber of the Vitrobot, a 3 µL sample was applied on a Quantifoil grid, which was surface plasma treated for 40 s at 5 mA just prior to use. Excess sample was removed by blotting using two filter papers for 3 s at −3 mm, and the thin film thus formed was plunged (acceleration about 3 g) into liquid ethane just above its freezing point. Vitrified films were transferred to the cryoholder and observed in the Tecnai Sphera microscope, at temperatures below −170 °C. Micrographs were recorded at low dose conditions, and at a magnification of 25,000 with typical defocus settings of 5 µm (Fig. 2a, b, Supplementary Fig. 3a–c) and 10 µm (Fig. 2c).

**STORM and TIRF microscopy.** STORM and TIRF images were acquired with a Nikon N-STORM system. Atto488 and Cy5 were excited using a 488 and 647 nm laser, respectively. Cy3 and Nile Red were excited using a 561 nm laser. Fluorescence was collected by means of a Nikon ×100, 1.4NA oil immersion objective and passed through a quad-band pass dichroic filter (97335 Nikon). Images were recorded with an EMCCD camera (ixon3, Andor, pixel size 0.17 µm). The movies were subsequently analyzed with the STORM module of the NIS element Nikon software. For STORM imaging (Fig. 2d), BTA polymers were prepared as stated above at a total BTA concentration of 50 µM with 5% BTA-Cy3, 25% BTA-DNA and 70% BTA-3OH. After equilibration overnight, the samples were diluted to 2 µM total BTA in TE/$Mg^{2+}$ buffer (10 mM Tris-HCl, 1 mM EDTA, 12.5 mM $MgCl_2$, pH 8) and mixed with 100 nM $R_E$, 100 nM $R_I$, 100 nM β-lactamase-Atto488 and 7.5 nM BLIP-Cy5. The sample was flown in a chamber between a glass coverslip (Menzel-Gläser, no. 1, $21 \times 26$ mm) and a glass slide which were separated by double-sided tape. After annealing for several minutes, the chamber was washed twice with TE-Mg buffer and twice with imaging buffer containing 50 mM Tris-HCl pH 7, an oxygen scavenging system (0.5 mg $mL^{-1}$ glucose oxidase, 40 µg $mL^{-1}$ catalase), 10% (w/v) glucose and 10 mM 2-aminoethanethiol. For TIRF imaging, BTA polymers were prepared as stated above and after equilibration for at least 1 day, 5% (relative to the total BTA concentration) Nile Red was added from a stock solution in methanol. After equilibration overnight, the samples were diluted

in PBS to 10 µM and flown in a chamber between a glass microscope coverslip and a glass slide as stated above.

**Enzyme activity assays.** For enzyme activity assays, samples were prepared in 384 well plates (Optiplate-384 Black, Perkin Elmer, cat. no.: 6007270) using a final sample volume of 50 µL in PBS supplemented with 1 mg $mL^{-1}$ bovine serum albumin. After the step-wise addition of each component, the samples were mixed by repetitive pipetting. In the final step, 5 µL substrate (CCF2-FA, Thermo Fisher, cat. no.: K1034) was added to a final concentration of 2 µM, the samples were mixed and the plates were spun down at 100 g for 1 min at room temperature. This resulted in a total dead time of about 2 min. The plates were then transferred to a Tecan Safire 2 Microplate Reader equilibrated to 25 °C. Fluorescence intensities were measured every 15 s using an excitation and emission wavelength of 410 nm and 447 nm, respectively.

**Fitting of the apparent inhibitory constant.** The apparent inhibitory constant $K_{i,app}$ between β-lactamase and BLIP upon recruitment on the BTA polymers was obtained by fitting Eq. 1 to the normalized enzyme activities as a function of inhibitor concentration shown in Fig. 3c, yielding a $K_{i,app}$ of $2.32 \pm 0.18$ nM.

$$V = V_b + V_0 \times \frac{K_{i,app}}{[I] + K_{i,app}} \qquad (1)$$

In Eq. 1, $V$ represents the measured normalized enzyme activity, $V_b$ the residual enzyme activity at saturating inhibitor concentrations, $V_0$ the normalized enzyme activity in absence of inhibitor minus the residual enzyme activity at saturating inhibitor concentrations (hence equals $1-V_b$), $[I]$ the inhibitor concentration and $K_{i,app}$ the fitted apparent inhibitor constant.

**Data availability.** The data that support the findings of this study are available from the corresponding authors upon reasonable request.

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

## Acknowledgements

We thank C.M.A. Leenders for the synthesis of BTA-DNA, N.M. Matsumoto for the synthesis of BTA-Cy3, and the ICMS animation studio for the graphics. This work was supported by funding from the Ministry of Education, Culture and Science (Gravity program, 024.001.035) and a European Research Council (ERC) Starting Grant (280255).

## Author contributions

S.P.W.W. and W.E. performed experiments, R.P.M.L. performed cryo-TEM imaging, E. W.M and M.M. supervised the research and provided advice, S.P.W.W., W.E., R.P.M.L., E.W.M., and M.M. analyzed the data and S.P.W.W., W.E., E.W.M., and M.M. wrote the manuscript.

## Additional information

**Competing interests:** The authors declare no competing financial interests.

