## [Peer Review File · Nature Communications]

Reviewers' comments:

Reviewer #1 (Remarks to the Author):

This exciting contribution describes the use of supramolecular polymers to recruit an enzyme and its inhibitor, and to influence the extent and rate of this process as compared to solution enzyme inhibition. Both enzyme and inhibitor proteins are attached to DNA strands. The supramolecular polymers are made from a mixture of benzene tricarboxamide BTA and BTA attached to a DNA strand. Addition of recruiting strands (10-mer hybridization) to the BTA/BTA-DNA polymer brings the two enzyme/inhibitor units in close proximity. The hybrid fibers were well characterized, and STORM verified that the system is not made of BTA-only fibers and free BTA-DNA. The authors chose an inhibition constant in the micromolar range and kept the solution concentrations in the nanomolar range. The supramolecular fibers are then expected to increase the effective molarities of both units, and were shown to significantly increase the inhibition. Importantly as well, BTA monomer exchange in the supramolecular polymer was shown to be significantly slower than the inhibition kinetics, suggesting that the rearrangement of enzyme and inhibitor along the fiber is mediated by fast detachment and reattachment of the 10-mer DNA strands. This contribution opens up a fascinating field of research, namely the interplay of dynamics in supramolecular systems with those of enzymatic reactions/inhibition, and the ability of one to influence the other. I therefore recommend acceptance of this manuscript, however the following comments need to be addressed:

1. can the authors comment on the fact that there seem to be fewer fibers in the TEM image when the enzyme and protein are added?
2. top of page 13: could the enzymatic activity also be faster than fiber formation? In other words, even before BTA-DNA forms a full fiber, a loose interaction of these structures and DNA hybridization brings the enzyme and inhibitor in closer proximity, in the same manner as other DNA-templated reactions? A control experiment where BTA-DNA is replaced by a DNA strand that does not form a fiber needs to be carried out.
3. In this manuscript, the authors chose to use a 10-base DNA cohesion, whose exchange turned out to be significantly faster than BTA exchange within the supramolecular polymer. It can then be argued that the kinetics are not a reflection of the polymer dynamics; that is, even if the strands were on a static structure, one would observe similar kinetics. In order to learn about the effect of polymer dynamics, a longer DNA strand cohesion should be used, so that DNA exchange is significantly slower than monomer exchange within the fiber. This experiment would significantly support the notion of using a dynamic supramolecular structure to influence enzyme activity.

Reviewer #2 (Remarks to the Author):

The work of Merks and Meijer describes the use of a linear supramolecular polymer based on BTA monomers as a dynamic platform for the recruitment of proteins, with consequent monitoring of their activity.

The dynamicity of BTA-based supramolecular polymers has been extensively studied by the authors' group and by the work of Brunsveld, where it was already demonstrated that proteins could be assembled and made to interact (by FRET) on the polymer backbone. In particular the dynamicity of the BTA systems lies in the fact that the hydrogen-bonded discoidal BTA monomers can exchange between the polymers via an insertion/deletion mechanism, either individually or in small clusters, where the polymer backbones/fibers possess weak points. Exchange of the monomers occurs over hours. Moreover, it was shown that functionalization of the BTA monomers (even with bulky units) does not considerably influence the exchange mechanism/time, demonstrating the robustness of the platform.

In this work, the authors synthesised a ssDNA-functionalised BTA monomer, complementary to recruiter ssDNA strands, which in turn are complementary either to a DNA-enzyme conjugate or a DNA-inhibitor conjugate. Polymer fibers containing different percentages of DNA-BTA monomers were successfully assembled and characterized by cryo-TEM and TIRF; likewise DNA-mediated recruitment of lactamase and BLIP on the polymer was confirmed by STORM using fluorescently tagged monomers and proteins. The lactamase/BLIP pair was created to have low affinity in solution, so that a pair would form only at high local concentrations, i.e. when recruited on the BTA polymer. Enzymatic activity on the polymer was then successfully controlled and monitored by different mechanisms:

- a) By increasing the inhibitor concentration in an active lactamase-BTA polymer
- b) By diluting the proteins-BTA monomer content in the polymer backbone using non-functionalised BTA monomers
- c) By adding DNA-BTA polymers to a mixture of DNA-BLIP and DNA-lactamase
- d) By mixing separately prepared BTA-BLIP and BTA-lactamase polymers
- e) Reversible removal and addition of lactamase from the polymer by using DNA toehold displacement

The use of the polymeric platform indeed increased the BLIP/lactamase complex formation by approx. 2600 times.

Overall, all the experiments and the work itself is done in a very meticulous way and is well presented. The authors also make the effort to provide a clear Supplementary Information with models to explain the increased enzyme affinity and some counter-intuitive results such as the 25% residual enzymatic activity at saturating inhibitor concentration. The literature is properly cited and I would add the following review:

W. Engelen, B. M. G. Janssen and M. Merkx, DNA-based control of protein activity, *Chem. Commun.* 2016, 52, 3598.

However, I would like the authors to consider the following points:

- In the introduction, the authors stress the importance of using dynamic system for controlling protein activity, such as reversible DNA-hybridisation and supramolecular polymers (BTA in this case). However, of the five mechanisms mentioned before, only b) specifically exploits the dynamicity of BTA, namely the monomer exchange, which occur in the time scale of hours. The other mechanisms all rely on the well-known reversible hybridization of DNA: in this work in fact, regulation of enzymatic activity occurs in minutes, much faster than the monomer exchange. As the authors suggest, regulation is most likely done by rapidly exchanging recruiter-DNA strands within the same polymer or even between different polymer chains.

- In this sense, a randomly and densely ssDNA-functionalized DNA origami or linear covalent polymer could in principle lead to the same results. In the time-scale of minutes, what is then the advantage of using a BTA supramolecular polymer in this specific case?

- The experiments of point c), d) and e) should be combined/followed by an experiment of type b), so that it can be shown that control of enzymatic activity can be exerted by both DNA hybridization and monomer exchange "at the same time".

- It would be nice to try to bring the monomer exchange reaction and the DNA hybridization to a similar time scale. How would the increase of a few nucleotides in the DNA handles affect the hybridization kinetics? It could also strengthen the hypothesis of the 25% unbound enzyme due to only partial hybridization.

With properly addressing and justifying the aforementioned points, I could see this manuscript being considered for publication.

Response to Reviewers' comments:

We are pleased that both reviewers appreciated our work and have used their feedback to further improve our manuscript. Please find a point-by-point response below, the changes in the text are highlighted in blue, while it is made visible in the annotated version of the revised version.

Reviewer 1

1. Can the authors comment on the fact that there seem to be fewer fibers in the TEM image when the enzyme and protein are added?

Because of variations in the thickness of the vitrified films used for Cryo-TEM (typically ~ 130 nm) and the high magnification (25000x), the number of fibers can vary substantially between different images of the same sample. Since the samples contained some 'contamination' with crystalline ice, we selected images that show a minimal amount of these ice particles. The TEM images are therefore not representative of the overall concentration of fibers.

2. Top of page 13: could the enzymatic activity also be faster than fiber formation? In other words, even before BTA-DNA forms a full fiber, a loose interaction of these structures and DNA hybridization brings the enzyme and inhibitor in closer proximity, in the same manner as other DNA-templated reactions? A control experiment where BTA-DNA is replaced by a DNA strand that does not form a fiber needs to be carried out.

We thank the reviewer for this comment. This question made us realize that the setup of the kinetic experiments was not described clearly enough. As described in Supplementary Information 1.6, all supramolecular BTA polymers were pre-assembled before adding them to other components (protein-DNA conjugates, recruiter strands). The BTA fibers are thus formed before they are added to the enzyme and inhibitor. *To avoid any confusion we have clarified this at several relevant places in the main text.* We also thank the reviewer for suggesting the additional control experiment in which the BTA-DNA strand is replaced by a DNA strand that does not form a fiber. *We have now performed this control experiment in two different ways:*

- 1) the BTA polymers were omitted and a 10 nt DNA strand was added identical to the handle on BTA-DNA
- 2) with BTA polymers consisting of 100% BTA-3OH in combination with the unconjugated 10 nt handle strand.

The latter experiment is the control experiment suggested by the reviewer and contains all the components that are present in the fully assembled system, except that the 10 nt DNA handle is not conjugated to the BTA. As expected neither of these controls show inhibition of enzyme activity, confirming that the presence of BTA polymers containing BTA-DNA is crucial for protein recruitment and subsequent complexation. These additional control experiments are now discussed in the main text and the experimental results have been added to the supporting information in Figure S4.

("Additionally, no inhibition of enzyme activity was observed when only the 10 nucleotide handle

strand was added, either in absence or presence of BTA polymers consisting of 100% BTA-3OH (Supplementary Figure 4).”

3. In this manuscript, the authors chose to use a 10-base DNA cohesion, whose exchange turned out to be significantly faster than BTA exchange within the supramolecular polymer. It can then be argued that the kinetics are not a reflection of the polymer dynamics; that is, even if the strands were on a static structure, one would observe similar kinetics. In order to learn about the effect of polymer dynamics, a longer DNA strand cohesion should be used, so that DNA exchange is significantly slower than monomer exchange within the fiber. This experiment would significantly support the notion of using a dynamic supramolecular structure to influence enzyme activity.

Our results indeed indicate that in our current system the rapid assembly of BTA-templated protein complex formation may be primarily determined by the rate of DNA-exchange. Based on these observations similar assembly kinetics may be anticipated on a static structure with a similar density of receptor strand, such as an origami surface or a covalent polymer. Whether this is the case remains to be demonstrated of course, because other aspects of the dynamics of the supramolecular polymer may still be important to allow efficient assembly. More importantly, even if exchange of BTA monomers is not directly involved in rapid protein complex formation, the self-assembling nature of the BTA-polymer still provides a crucial advantage compared to more static structures such as DNA-origami surfaces or covalent polymers. As demonstrated in Figure 3, the non-covalent assembly of the BTA-template allows precise and easy tuning of the density of DNA-handles on the BTA polymer, which through its effect on the effective local concentration provides quantitative control over the extend of protein complex formation. We realize that this aspect did not receive sufficient attention in the original manuscript and therefore now address this important advantage of a dynamic self-assembling platform both in the introduction and in the discussion. (*“Supramolecular assembly allows straightforward tuning of the density of functional BTA monomers within these BTA polymers and previous work has also shown that the distribution of functional monomers can be externally controlled, e.g. via interaction with multivalent templates” and “Moreover, the supramolecular nature of the BTA platform allows one to efficiently tune the density of DNA-handles on the BTA polymer, which in turn allows precise control of the effective local concentration and optimization of the system for different protein complexes.”*)

While the specific combination of relatively slow supramolecular polymer dynamics and fast DNA hybridization kinetics observed in this work may already be optimal for many applications, we agree it would be very interesting to study the behavior of the system when the kinetics of these two dynamic processes are more similar. This could be achieved by using longer DNA strands to slow down DNA-exchange as suggested by the reviewer, but also by changing the properties of the BTA-scaffold, e.g. by increasing the exchange rate of BTA-DNA monomers. This important aspect for future work is now discussed more explicitly in the discussion. (*“The 10-nucleotide DNA-receptor strand used in this study thus provides sufficient thermodynamic driving force to ensure efficient recruitment of low nanomolar concentrations of proteins on the BTA-polymer, while at the same time allowing rapid exchange of DNA-conjugated proteins along the BTA-scaffold. A more thorough understanding of the importance of supramolecular polymer dynamics and the kinetics of DNA*

hybridization may be obtained by tuning the length of the DNA receptor strand, and/or by changing the exchange kinetics of the BTA monomer.”)

Reviewer #2

Overall, all the experiments and the work itself is done in a very meticulous way and is well presented. The authors also make the effort to provide a clear Supplementary Information with models to explain the increased enzyme affinity and some counter-intuitive results such as the 25% residual enzymatic activity at saturating inhibitor concentration. The literature is properly cited and I would add the following review:

W. Engelen, B. M. G. Janssen and M. Merckx, DNA-based control of protein activity, Chem. Commun. 2016, 52, 3598.

We thank the reviewer for the suggestion and we have included the suggested review paper as an additional reference in the main text.

However, I would like the authors to consider the following points:

1. In the introduction, the authors stress the importance of using dynamic system for controlling protein activity, such as reversible DNA-hybridisation and supramolecular polymers (BTA in this case). However, of the five mechanisms mentioned before, only b) specifically exploits the dynamicity of BTA, namely the monomer exchange, which occur in the time scale of hours. The other mechanisms all rely on the well-known reversible hybridization of DNA: in this work in fact, regulation of enzymatic activity occurs in minutes, much faster than the monomer exchange. As the authors suggest, regulation is most likely done by rapidly exchanging recruiter-DNA strands within the same polymer or even between different polymer chains.

In this sense, a randomly and densely ssDNA-functionalized DNA origami or linear covalent polymer could in principle lead to the same results. In the time-scale of minutes, what is then the advantage of using a BTA supramolecular polymer in this specific case?

The non-covalent nature of the supramolecular polymeric platform allows the modular combination of different monomers providing direct control over the polymer composition. This enabled us for example to easily produce polymers with different ratios of BTA-3OH and BTA-DNA providing a range of BTA-DNA densities as used for the experiments related to figure 3d. This would take significantly more effort when working with covalent platforms such as covalent polymers and DNA-origami as this would require synthesis and characterization of each new variant. See also our more extensive response to question 3 of reviewer 1.

2. The experiments of point c), d) and e) should be combined/followed by an experiment of type b), so that it can be shown that control of enzymatic activity can be exerted by both DNA hybridization and monomer exchange “at the same time”.

The sequential control of enzymatic activity by DNA hybridization followed by BTA monomer exchange, which the reviewer proposes, was actually already shown in figure 4, but we realize that we should have explained this more clearly. In the experiment shown in Figure 4b and c (mechanism c and d in the reviewer's overview) the final state (reached within minutes) represents a polymer with both the enzyme and inhibitor protein recruited, hence the enzyme is inhibited. The initial state in Figure 4a as well as the control shown as the red diamond also represent a state where the enzyme and inhibitor are recruited and complexed on the supramolecular polymer. During the experiment in figure 4a (mechanism b in the reviewer's overview), we then subsequently added additional BTA-OH to result in a final BTA-DNA concentration of 0.25%, which resulted in the reversion of enzyme inhibition over a period of several hours due to dilution of the BTA-DNA monomers within the supramolecular polymer. We have included a comprehensive description of these different steps in Figure legend 4. (*"The green diamonds represent the activity of a sample without inhibitor. The red diamonds represent the activity of the system containing 5% BTA-DNA, which was not diluted by addition of 100% BTA-3OH."*)

3. It would be nice to try to bring the monomer exchange reaction and the DNA hybridization to a similar time scale. How would the increase of a few nucleotides in the DNA handles affect the hybridization kinetics? It could also strengthen the hypothesis of the 25% unbound enzyme due to only partial hybridization.

See our response to point 3 of Reviewer 1. In short, we agree that it would be interesting to make the monomer exchange rates and the kinetics of DNA hybridization kinetics more similar. This could be done by increasing the length of the DNA handles by a few nucleotides, as suggested by the reviewer, or by tuning the properties of the BTA-monomer, e.g. the length of the hydrophobic alkyl chain. These and some of the many other possibilities to tune and subsequently exploit the remarkable properties of this modular self-assembling system will be explored in future studies.

Finally, we carefully checked the compliance of our manuscript with the manuscript checklist. As a result we have moved the description of the experimental methods from the Supplementary Information to the main text. In addition, we changed the coloring scheme used in Figures 1, 4, 5 and S5 from red/green to magenta/turquoise. We hope that with these improvements the manuscript can now be accepted for publication.

REVIEWERS' COMMENTS:

Reviewer #1 (Remarks to the Author):

This reviewer is satisfied with the revised manuscript and authors' responses, and recommends acceptance of this paper.

Reviewer #2 (Remarks to the Author):

I thank the authors for the responses to the comments, now it is clear the the BTA-3OH dilution experiment was basically performed after the other mechanism of control.

The only thing that leaves me a a bit bothered is not seeing as mentioned full control over the system in terms of kinetics, as this would be a real novelty and would completely exploit the dynamic nature of this extensively studied polymer and DNA hybridisation. I also understand that synthesising longer DNA strands and repeat all the experiments would require time.

Hopefully as the authors mention, this will be the topic of future works...